# Evaluation of annual maximum snow depth data estimation from the European-wide reanalysis C3S MTMSI (Copernicus Climate Change Service - Mountain Tourism Meteorological and Snow Indicators) against in-situ observations

Elisa Kamir[1,2], Samuel Morin[1], Guillaume Evin[2], Penelope Gehring[3], Bodo Wichura[3], and Ali Nadir Arslan[4,5]

[1]Météo-France, CNRS, Univ. Grenoble Alpes, Université de Toulouse, CNRM, Centre d'Études de la Neige, Grenoble and Toulouse, France
[2]Univ. Grenoble Alpes, INRAE, CNRS, IRD, Grenoble INP, IGE, Grenoble, France
[3]Deutscher Wetterdienst, Regionales Klimabüro Potsdam, Germany
[4]Arctic Space Centre, Finnish Meteorological Institute, Helsinki, Finland
[5]School of Technology and Innovations, University of Vaasa, Vaasa, Finland

**Correspondence:** Samuel Morin (samuel.morin@meteo.fr)

**Abstract.** Large snow load events are a major hazard for both human societies, in particular buildings and transport safety, and natural ecosystems. National and European frameworks provide guidelines and standards in order to take into account extreme snow load hazard in infrastructure design. However, there is a lack of reference data for their implementation. This is even more challenging in the context of climate change, which modifies the frequency and intensity of major snow load events. In the context of the Framework Partnership Agreement on Copernicus User Uptake, we have developed a pan-European extreme value analysis of annual snow load maxima based on the Mountain Tourism Meteorological and Snow Indicators (MTMSI) dataset available from the Copernicus Climate Change Service. This dataset includes reanalysis data for the period 1962-2015, based on the UERRA (Uncertainties in Ensembles of Regional Reanalyses) reanalysis and snow cover simulations, and past and future climate projections based on regional climate simulations for the period 1951-2100. Here, we describe the evaluation of the MTMSI reanalysis component in terms of annual snow depth maxima against multiple in-situ observation datasets. Results are provided at the NUTS-3 (Nomenclature des unités territoriales statistiques) scale used in MTMSI, for multiple elevations over a large area stretching from the European Alps to the Scandinavian countries. For 75% of the comparisons between observed and simulated snow depth maxima, we report absolute bias scores between -0.23 m and 0.15 m, correlations above 0.59, and a Kling-Gupta Efficiency metric above 0.29. We identify some areas where MTMSI does not adequately portray in-situ observations of snow depth maxima, located in the Alps and coastal areas of the Netherlands, Norway, Sweden, and Croatia. This study thus provides background information for assessing the relevance of this pan-European dataset in terms of annual snow depth maxima, relevant for annual snow mass and snow load maxima based on complementary information derived from snow cover model output. The MTMSI annual maximum snow depth reanalysis dataset is available through the following link: https://doi.org/10.5281/zenodo.15181401 (Kamir et al., 2025).

# 1 Introduction

Large amounts of snow, due to extreme snowfall events or the accumulation of snow from multiple snowfall events during a given time period, are a challenging hazard for transportation and housing infrastructure. Snow load, i.e. the pressure exerted by the snowpack on underlying surfaces (in $N\,m^{-2}$ or Pa), is directly calculated from the snow mass (also referred to as the snow water equivalent, SWE, expressed in $kg\,m^{-2}$ of mm water equivalent), multiplied by the gravitational acceleration constant $(g = 9,8\,m\,s^{-2})$. National and European frameworks provide guidelines and standards in order to take into account extreme snow load hazard in infrastructure design. However, as shown by Croce et al. (2019), there is a lack of harmonization between the different countries, and, in particular, the reference observations used to estimate extreme snow loads vary in quantity and quality. Furthermore, estimating extreme snow loads is even more challenging in the context of climate change, which influences the frequency and intensity of major snow load events (Croce et al., 2021).

This study relies on an ensemble of snow projections derived from the Mountain Tourism Meteorological and Snow Indicators (MTMSI) dataset (Morin et al., 2021). The MTMSI dataset is available at the scale of NUTS-3 regions (Nomenclature des Unités Territoriales Statistiques, EUROSTAT, 2015) which correspond to administratively relevant regions defined for each country across Europe. The MTMSI reanalysis dataset was evaluated in several recent studies. Morin et al. (2021) compared MTMSI reanalysis data, in terms of the number of days with SWE values above $100\,kg\,m^{-2}$, for the Savoie NUTS-3 region (FRK27), at various elevations, with the results of the SAFRAN - SURFEX/ISBA-Crocus - MEPRA (S2M) reanalysis (Vernay et al., 2022). They showed that, in many cases, MTMSI and S2M data followed similar patterns. However, deviations were found in particular at high elevation, where MTMSI values were generally lower than S2M. These deviations were identified to be consistent with the lower resolution of the UERRA dataset (5.5 km), and the low density of precipitation observation networks at high elevation often associated with an underestimation of solid precipitation due to, e.g., undercatch (Kochendorfer et al., 2020). More recently, Monteiro and Morin (2023) have also evaluated the MTMSI reanalysis dataset based on in-situ observations and remote sensing snow cover fraction datasets across the European Alps domain, as well as various reference temperature and precipitation datasets. They assessed monthly and seasonal snow cover variables (snow depth and snow cover duration) and their main atmospheric drivers (near-surface temperature and precipitation). Monthly snow depth values lead to mean absolute errors generally below 30%, and correlation values close to 0.9 at all elevations. However, to date, extreme snow variables (e.g. snow depth or SWE annual maxima) used in snow load assessments were not available at the pan-European scale. These variables were simulated as part of the MTMSI reanalysis but not made available so far.

The main goal of the present study is to provide an evaluation of the MTMSI reanalysis in terms of maximum annual snow depth values at the European scale, against verified in-situ observations. Although the focus of extreme snow load analysis would ideally require using SWE observation datasets for the evaluation, there are far more snow depth observations than SWE observations, and the Crocus snow cover model has been shown to simulate equally well snow depth and SWE (Krinner et al., 2018). The performance of the dataset in terms of snow depth should thus translate in terms of SWE. For the present study, we gathered and combined various in-situ snow depth datasets in order to evaluate how MTMSI reanalysis snow depth values compare to in-situ snow depth observations. We further aim at assessing the quality of this dataset as a function of the

geographic location and the elevation (in mountain areas). This analysis focuses exclusively on annual snow depth maxima
from 1962 through 2015. We carried out the evaluation on a horizontal and vertical scale, as some NUTS-3 regions are covered
by the MTMSI dataset on multiple elevation scales.

Section 2 describes the data and methods used in this study, in particular the MTMSI dataset and the snow depth observations.
Section 3 details the results of this evaluation. Section 4 further discusses the main strengths and limitations of the MTMSI
dataset concerning extreme snow variables. Section 5 concludes.

## 2 Study area, material and method

### 2.1 Study area

The evaluated MTMSI reanalysis dataset covers a pan-European domain. It includes the European Union, candidate countries,
and members of the European Free Trade Association. Altogether, the dataset covers EU member states, Albania, Andorra,
Montenegro, North Macedonia, Serbia, Turkey, the United Kingdom, Iceland, Liechtenstein, Norway, and Switzerland (Morin
et al., 2021). Figure 1 shows the mean elevation of the NUTS-3 regions (administratively relevant regions defined for each
country across Europe) across this pan-European domain and highlights the regions with high elevations (e.g. the Alps, east
of Turkey). For "plain" NUTS-3 regions, snow simulations are only available at their mean elevation (NUTS-3), while snow
simulations are provided by 100-meter elevation band for "mountain" NUTS-3 regions.

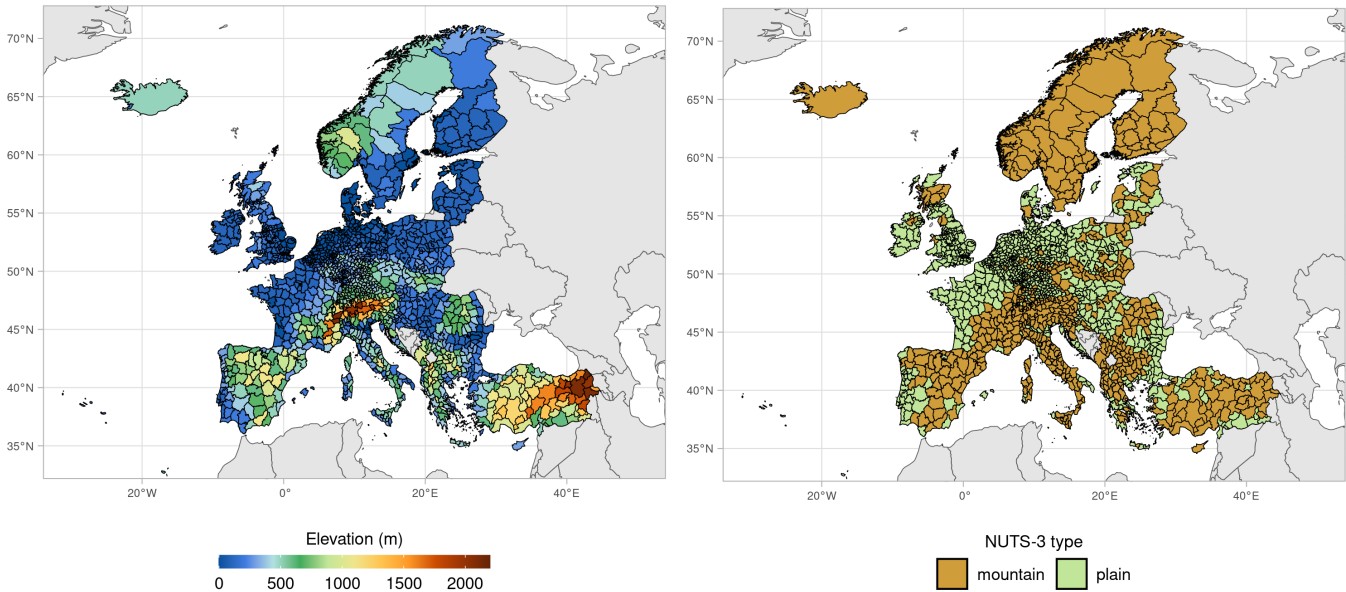

**Figure 1.** Mean elevation (left) and type (right) of each NUTS-3 (Nomenclature des Unités Territoriales Statistiques) regions of the MTMSI dataset. Representative grid points from UERRA were based on case-by-case identification of "mountain" and "plain" NUTS-3. UERRA grid points were selected to provide results by steps of 100 meters elevation on "mountain" NUTS-3 regions, while for each "plain" NUTS-3 region, one UERRA grid point was selected to represent the average elevation of the NUTS-3 region.

## 2.2 Data

### 2.2.1 MTMSI reanalysis dataset

This study evaluates annual snow depth maxima from the Copernicus Climate Change Service (C3S) Mountain Tourism Meteorological and Snow Indicators (MTMSI) against in-situ snow depth observations across Europe. The full description of the dataset is provided by Morin et al. (2021), and briefly summarized here. This reanalysis is derived from the UERRA reanalysis, which spans the time period from 1961 through 2015 in Europe, at 5.5 km horizontal resolution (Soci et al., 2016). For MTMSI, only a geographic subset of the full UERRA pan-European reanalysis was used, in order to operate over a tractable number of locations (the UERRA reanalysis includes 1,142,761 grid points). The primary geographical unit components of the MTMSI dataset are NUTS-3 regions, corresponding to administratively relevant regions defined for each country across Europe. In mountainous areas, UERRA reanalysis grid points were selected to represent the meteorological conditions in each NUTS-3 region by steps of 100 m elevation. In non-mountainous areas, one UERRA reanalysis grid point was selected to obtain the optimal agreement with the mean elevation of the NUTS-3 region (Figure 1). The near-surface atmospheric fields of the reanalysis were used to drive the snow cover model Crocus, part of the SURFEX/ISBA land surface model (Vionnet et al., 2012; Masson et al., 2013). This provides daily snow depth and snow water equivalent data for each of the 5625 mountain locations

(NUTS-3/elevation pairs) and 959 plain locations (at the mean elevation of the corresponding NUTS-3 region), totaling 6584 NUTS-3/elevation pairs. This subset corresponds to 0.5% of the grid points of the UERRA reanalysis.

From these daily data, including daily atmospheric variables (temperature, precipitation), 39 indicators were computed to form the MTMSI dataset available from C3S (Morin et al., 2021). However, the annual snow depth or SWE maximum is not included among the 39 indicators of this dataset, which has primarily been developed in relation to the mountain tourism sector (Morin et al., 2021; François et al., 2023). Therefore, the annual maximum snow depth values were computed based on the daily data. The maximum was computed for each year N from August 1st, N-1 to July 31st, N. The resulting dataset of annual

maxima of snow depth is available at https://doi.org/10.5281/zenodo.15181401.

### 2.2.2   In-situ observation datasets

We evaluated the MTMSI dataset based on in-situ data whose characteristics are described below, and summarized in Table 1. We gathered observation data from multiple countries and sources, in order to evaluate the MTMSI dataset in a variety of climates and landforms (see Figure 2). Since some in-situ stations appeared in two datasets simultaneously, we removed the

duplicates when merging the four datasets. Table 1 introduces the characteristics of the subsetted in-situ datasets, i.e. after duplicated stations were removed. All data underwent quality checking performed by data providers. Most stations provided data on a time window of less than 30 years during the period 1962-2015.

| Location | Number | Time scale | Reference |
| --- | --- | --- | --- |
| European Alps | 2861 stations | Daily at variable time | (Monteiro and Morin, 2023) |
| Germany | 4930 stations | Daily at 6 UTC | (Castino and Wichura, 2020) |
| Finland | 33 stations | Daily morning and evening measure at variable time | World Meteorological Organization recommendations (WMO, 2021) |
| Northern / Eastern Europe | 5877 stations | Daily mean | (Klein Tank et al., 2002; Klok and Klein Tank, 2009) |

**Table 1.** Characteristics of snow depth in-situ observation datasets used for the study.

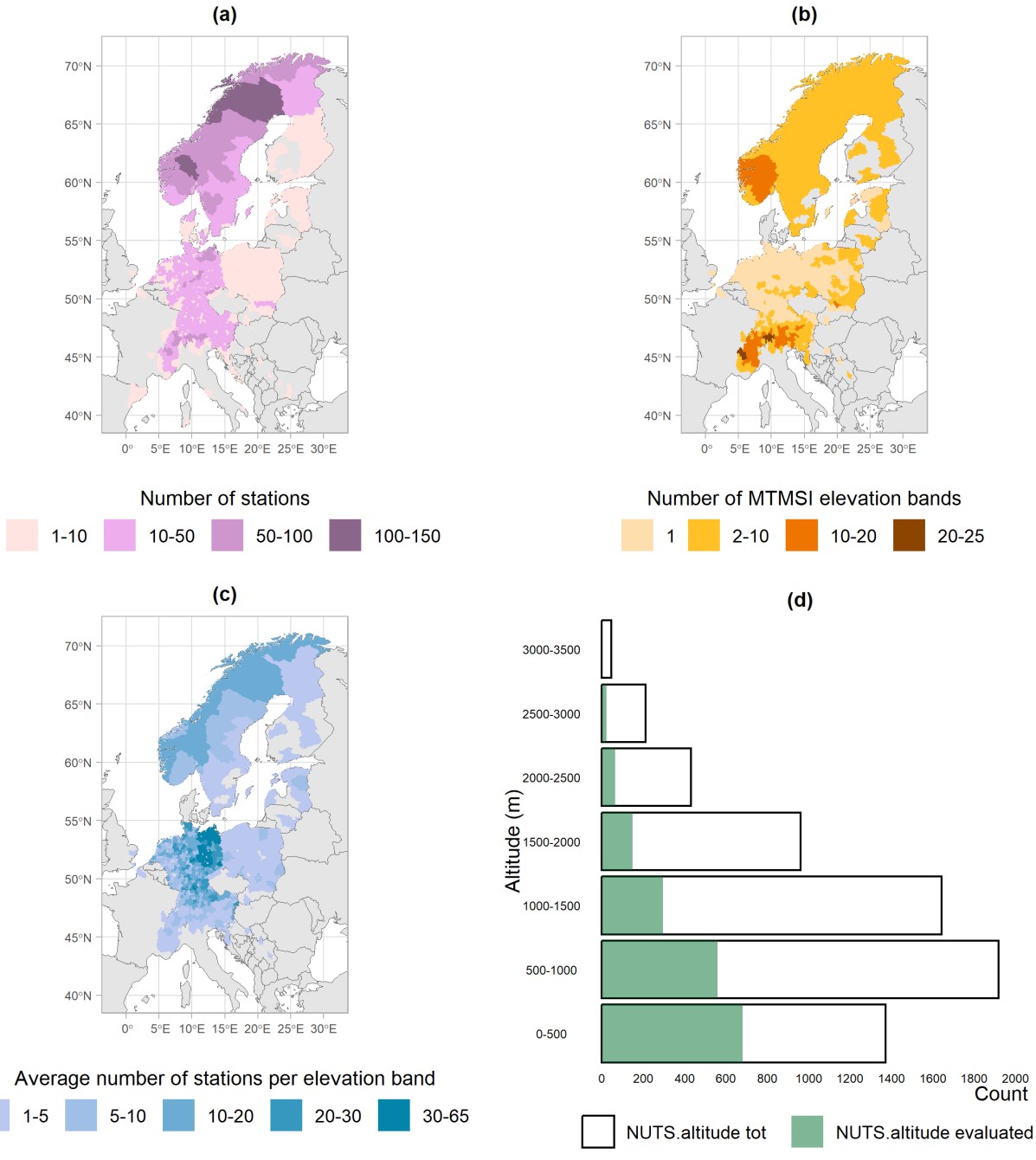

**Figure 2.** Characteristics of the observation data set: (a) number of snow depth observation stations; (b) number of MTMSI elevation bands evaluated based on observation stations; (c) mean number of stations per MTMSI elevation band used for the evaluation; (d) fraction of the MTMSI dataset that has been evaluated against at least one observation station, per 500 m elevation range. MTMSI evaluation based on largest numbers of stations are located in the European Alps region, Germany and Scandinavian countries.

### Dataset covering the European Alps

This dataset covering the European Alps is an ensemble of daily in-situ observations from 2861 stations spanning the 1961–2019 period. It contains daily snow measurements from stations covering Austria, France, Germany, Italy, Slovenia, and Switzerland, and has been published by Matiu et al. (2021). The largest part of the data results from manual measurements. Data quality was ensured by Matiu et al. (2021) through the following processes: (1) values below zero were set to missing values; (2) false null values were investigated by checking their spatial consistency; (3) temporal consistency was checked (series with jumps larger than 50 cm on two consecutive days were manually checked); (4) series that showed a "dubious" behaviour were marked (i.e. inconsistency between snow depth and depth of snowfall, unlikely values, improbable temporal variability, multiple seasons with no snow, or excessive gaps); (5) general spatial consistency was checked.

### Dataset covering Germany

The dataset provided by the German meteorological service (DWD) contains daily snow depth measurements from 5496 stations for each winter season (September–May) from 1950 through 2020. Although these measurements are also freely accessible through the DWD data portal at https://www.dwd.de/EN/ourservices/cdc/cdc_ueberblick-klimadaten_en.html, the data provided have undergone additional quality checks to ensure higher data reliability. The data verification was carried out by DWD scientists (Castino and Wichura, 2025) and included: (1) isolated null values have been validated by checking their consistencies with nearby stations; (2) in a similar way, long series of null values were investigated. Indeed, these series sometimes result from a mismatch between data and metadata (starting date), causing false null values to be set instead of missing values; (3) the consistency of the temporal variability of snow depth measurements was evaluated by comparing with the air temperature and precipitation variability; (4) the homogeneity of the time series was evaluated by performing the standard normal homogeneity test (SNHT). We removed stations that were already in the "European Alps" dataset (see above). Hence, we used a subset of this dataset containing measurements from 4930 stations.

### Dataset covering Finland

The dataset provided by the Finnish Meteorological Institute (FMI) contains sub-daily timeseries of snow depth on 1961-2015, for 33 stations covering Finland. Data quality was ensured by automatic control tests, conducted by the FMI, that are divided into 3 categories: jump tests, threshold tests, and growth-rate tests. The first ones check the consistency of snow depth variations between observations separated by 10 minutes. The second ones verify that the maximum snow depth values are not above a threshold that depends on the latitude. The last ones check the consistency of snow depth variations between observations separated by 60 minutes in regard to other atmospheric variables. Each observation is stamped with a flag depending on which tests it passes and is either directly validated or goes through human quality control, in which a meteorologist analyzes the observation and either accepts or disqualifies it. As two measures were available each day, we picked the morning one, for which the time was not strictly the same from one day to another.

### Dataset covering Northern and Eastern Europe

The dataset provides mean daily snow depth for about 9777 stations spread over Scandinavian countries and eastern countries of Europe, spanning from Turkey to Poland and the Netherlands (see Figure 2). The data is gathered by the European Climate Assessment Dataset (ECA&D) and is freely accessible through its website (https://www.ecad.eu). Whenever synoptical snow

depth data is available at 00, 06, 12 and/or 18 UTC, ECA&D scientists calculated the daily mean snow depth as the average snow depth of the available values. Data quality is ensured by the authors by performing automated and manual checking (Klein Tank et al., 2002; Klok and Klein Tank, 2009). Automated procedures ensure that data is always above 0 cm, and (1) if the station is below 400 m, the measured value is below 300 cm; (2) if the station is between 400 m and 2000 m, the measured value is below 800 cm; (3) if the station is above 2000 m, the measured value is below 1500 cm (ECA&D, 2013). If a requirement is not met, the value can be manually checked. A resulting quality code is provided along with the data. Only data tagged as "valid" were kept for this analysis. A further quality check made us remove the data from three stations that appeared dubious, in Norway and Croatia. We removed stations that were already in the "European Alps" or "Germany" dataset (see above), hence, we used a subset of this dataset containing measurements from 5877 stations.

**Merged observation dataset**

We gathered all of the observation sets together. We filtered out identical stations. As we conducted the evaluation on the MTMSI dataset time span (from 1962 through 2015), we cropped the in-situ datasets from 1961-08-01 through 2015-07-31. Then, we removed stations that do not provide any complete winter, defined as less than 10% daily data missing between the 1st October and the 30th of April. In the end, our dataset consists of 11,782 stations providing at least one winter maximum from 1962 through 2015. Note that the adjustment of EURO-CORDEX climate simulation data uses MTMSI atmospheric data from 1980 through 2012 (Morin et al., 2021), which includes the evaluation period considered here.

The evaluation is restricted to the coverage of observation datasets, which do not cover the whole area: Mediterranean countries, as well as some eastern countries, are therefore missing in this evaluation (Figure 2).

## 2.3 Method

In this section, we describe how we compared each NUTS-3/elevation snow depth maximum values to the observations, noting that for each NUTS-3/elevation pair there can be numerous observation stations. We aimed at evaluating as much of the reference NUTS-3/elevation data as possible, both in plain and mountain areas.

### 2.3.1 Screening the length of station time series

We only computed the annual snow depth maximum if the records had less than 10% daily data missing between the 1st of October and the 30th of April, in order to reduce the probability of missing the actual snow depth maxima. Based on this analysis, we filtered out 3,178 station time series providing less than 20 full winters, leaving 8,604 stations with at least 20 full winters.

### 2.3.2 Spatial pairing of each MTMSI and observation

The MTMSI dataset provides data for 1,517 NUTS-3 regions. In order to conduct the evaluation of each individual NUTS-3/elevation, we selected the stations located within the NUTS-3 regions, and at the same elevation - with a 150 m tolerance. In that respect, 831 stations were discarded as they could not be paired with any individual NUTS-3/elevation.

Given that for mountain NUTS-3 regions MTMSI data are provided by steps of 100 m, a given observation station can thus contribute to the evaluation of several NUTS-3/elevation pairs of the MTMSI dataset (Figure 2). Based on the availability of the observation stations and the match with the MTMSI geometry, we were able to assess MTMSI annual snow depth maxima for 671 NUTS-3 regions and 1,771 NUTS-3/elevation pairs. Among the 8,604 stations that are long enough, 7,773 time series of observed annual snow depth maxima match the MTMSI dataset, i.e. the NUTS-3/elevation pairs. Indeed, as indicated above, MTMSI only provides snow data at the mean elevation of plain NUTS-3 regions. For these NUTS-3 regions, stations are often located either below or above this mean elevation. At the end, the evaluation was carried out using 7,773 time series of observed annual snow depth maxima. As indicated above, a given time series of observed maxima can be used to evaluate several NUTS-3/elevation MTMSI time series, and this evaluation comprises 12,350 comparisons between MTMSI and observed maxima.

Figure 2d provides the distribution of this total number by 500-meter elevation range (in green) and illustrates a higher availability of observations at low elevations. Most of the evaluation was carried out for NUTS-3/elevation pairs at elevations below 1500 m - where most observation stations are located - and they represent 31% of the MTMSI dataset below 1500 m. In total, 27% of the MTMSI dataset has been evaluated against at least one observation data (47% of plain NUTS-3 regions and 23% of mountain NUTS-3 regions).

### 2.3.3 Statistical scores

The evaluation of MTMSI snow depth annual maxima was performed by comparing their values against observed snow depth maxima on the period 1962-2015, using several statistical scores, for each of the NUTS-3/elevation pairs. Each NUTS-3/elevation pair could have no station, one station, or more. Thus, the number of statistical scores available depends on each NUTS-3/elevation pair. Let $y_i$, for $i = 1, \ldots, N$ denote the snow depth annual maxima in the MTMSI dataset for a NUTS-3/elevation pair, and $x_i$, for $i = 1, \ldots, N$ denote the observed snow maxima from a selected station, where $N$ is the number of full winters in the period 1962-2015 for this station. The following scores are considered:

– **Bias:**

$$b = \frac{\sum_{i=1}^{N}(y_i - x_i)}{N}.$$

– **Pearson's correlation**:

$$r = \frac{\sum_{i=1}^{N}(x_i - \bar{x})(y_i - \bar{y})}{\sqrt{\sum_{i=1}^{N}(x_i - \bar{x})^2}\sqrt{\sum_{i=1}^{N}(y_i - \bar{y})^2}},$$

where $\bar{x}$ and $\bar{y}$ are the mean of the observed and MTMSI snow maxima, respectively. In order to characterize the variability of the Pearson's correlations for different stations associated to a NUTS-3/elevation pair, we also compute the median value of these Pearson's correlations for each NUTS-3/elevation pair.

– **Modified Kling-Gupta Efficiency metric (Gupta et al., 2009; Kling et al., 2012)**:

$$KGE = 1 - \sqrt{(r-1)^2 + (\beta - 1)^2 + (\gamma - 1)^2},$$

where $\beta = \frac{\bar{y}}{\bar{x}}$, and $\gamma = \frac{\sigma_y/\bar{y}}{\sigma_x/\bar{x}}$ where $\sigma_x$ and $\sigma_y$ are the standard deviations of MTMSI and observed snow depth maxima, respectively.

The KGE score expresses, in a single value, the differences between the MTMSI snow depth maxima and the observed ones. Initially developed to assess hydrological model performances, it combines the correlation $r$, bias ratio $\beta$, and variability ratio $\gamma$. It has shown great added value for assessing precipitation from reanalysis (Gomis-Cebolla et al., 2023) or satellite (Baez-Villanueva et al., 2018). KGE values greater than -0.41 indicate that a model improves upon the mean (Knoben et al., 2019), which would mean in our case that MTMSI provides a better estimate of snow depth annual extremes than the climatological values derived from the observations.

## 3 Results

This section provides statistical scores characterizing the performance of the MTMSI annual snow depth maxima against in-situ observed snow depth maxima across Europe, from 7,773 stations with at least 20 annual maxima from 1962 through 2015 and matching MTMSI geometry.

### 3.1 Illustration for two NUTS-3 region / elevation pairs

Figure 3 shows the time series of annual snow depth maxima from MTMSI and in-situ stations at 500 m on NUTS-3 region 'DEA5A' (located in Nordrhein-Westfalen region from Germany), and at 1700 m on NUTS-3 region 'FRL03' (Alpes Maritimes, located in the southern French Alps). For the NUTS-3 region 'DEA5A', we can see that 12 in-situ stations were selected for the evaluation (Figure 3a), and that they are matching MTMSI time series with a satisfactory correlation. Indeed, the median bias value is $6.1 \times 10^{-3}$ m and the median correlation is 0.84. For the NUTS-3 region 'FRL03', only 2 in-situ stations were selected for the evaluation (Figure 3b), but they are following the same variations as MTMSI annual snow depth maxima, with a close agreement. The median bias value is -0.13 m and the median correlation is 0.9. These two examples also show that the number of in-situ stations used for the evaluation of MTMSI is variable, and that a limited number of stations does not necessarily degrade the evaluation scores.

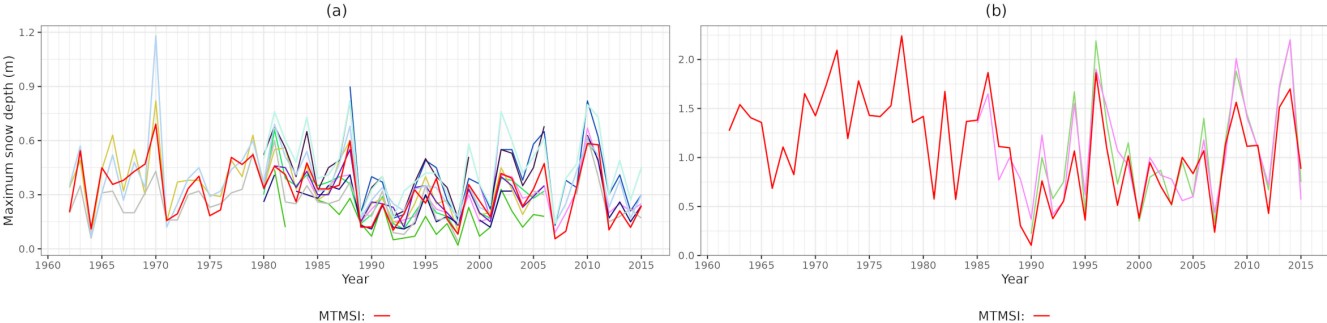

**Figure 3.** Time series of annual snow depth maxima from MTMSI and in-situ stations (a) at 500 m on NUTS-3 region 'DEA5A' (located in Nordrhein-Westfalen region from Germany), and (b) at 1700 m on NUTS-3 region 'FRL03' located in the southern French Alps.

## 3.2 MTMSI dataset scores on mean elevation of European NUTS-3 regions

Table 2 provides a summary of the scores obtained for 12,350 comparisons between observed and simulated snow depth
maxima. As indicated above, these comparisons cover 671 NUTS-3 regions, evaluate 1,771 NUTS-3/elevation pairs, and span the period 1962 - 2015. Median scores (i.e. the 50-% quantiles) indicate that for 50% of these comparisons, KGE and correlation values exceed 0.51 and 0.70, respectively. Median relative and absolute biases are equal to -8.9% and -0.02 m, which indicates that MTSMI maxima tend to slightly underestimate observed maxima overall. For 95% of the comparisons (i.e. above the 5% quantiles), KGE and correlation scores exceed -0.37 and 0.34, respectively, which means that MTMSI maxima perform better
than the mean observed maxima in at least 95% of the cases. For 75% of the comparisons (differences between 12.5% and 87.5% quantiles), absolute biases are between -0.23 m and 0.15 m, and between -40% and 36% in terms of relative biases.

| Quantile | KGE | Relative bias | Absolute bias | Correlation |
|---|---|---|---|---|
| **5%** | -0.37 | -53.8% | -0.42 m | 0.34 |
| **12.5%** | 0.06 | -40.4% | -0.23 m | 0.49 |
| **25%** | 0.29 | -28.3% | -0.09 m | 0.59 |
| **50%** | 0.51 | -8.9% | -0.02 m | 0.70 |
| **75%** | 0.64 | 14.8% | 0.04 m | 0.78 |
| **87.5%** | 0.70 | 36.0% | 0.15 m | 0.82 |
| **95%** | 0.76 | 68.1% | 0.37 m | 0.86 |

**Table 2.** Evaluation scores distribution main quantiles. Each score's value population includes 12,350 comparisons between MTMSI and observed maxima, evaluating 1,771 NUTS-3/elevation pairs in total.

### 3.2.1 Kling Gupta Efficiency

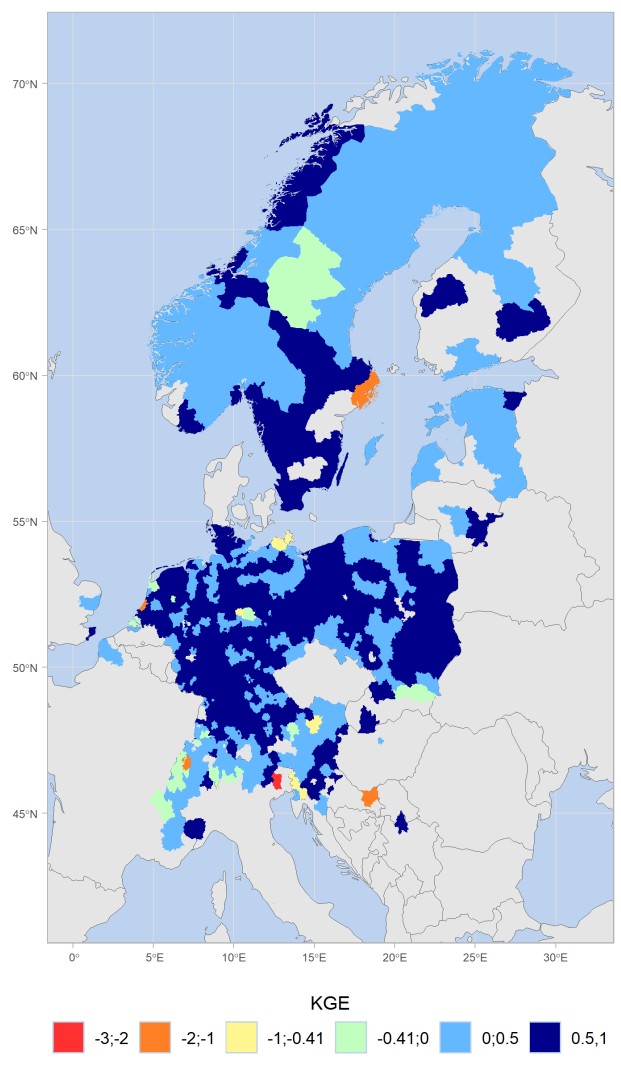

**Figure 4.** Modified Kling Gupta Efficiency (KGE) metric on each NUTS-3 region's mean elevation resulting from MTMSI evaluation.

Figure 4 displays the value of KGE for 646 NUTS-3 regions at their mean elevation, out of the 671 NUTS-3 regions which are evaluated (meaning that 25 NUTS-3 regions did not have enough observed data close to their mean elevation). These 646 KGE scores are derived from the median KGE values obtained from the different stations corresponding to these NUTS-3/elevation pairs, and summarize MTMSI skills in various climates, from Scandinavian countries to some Mediterranean regions. 635 NUTS-3 regions (i.e. 98%) show KGE values above -0.41 at their mean elevation, which defines the skill threshold indicating better skills of MTMSI compared to climatological statistics. 75% of the NUTS-3 regions have KGE values above

0.45, indicating satisfactory skills of MTMSI at the mean elevation of those 646 NUTS-3 regions. The NUTS-3 regions from
Germany and its Eastern regions, where elevation is spanning from 0 m to 500 m, show the highest KGE values, reaching
up to 0.84 for the NUTS-3 region 'DEC03' in Saarland region of Germany. Lower values of KGE are obtained for NUTS-3
regions from the European Alps, where the mean elevations are higher (from about 1000 m). Scores are still satisfactory at
these elevations for some NUTS-3 regions, like the 'AT341' from Vorarlberg region of Austria reaching 0.68 at 1600 m. Some
low and negative outlying values are displayed for a few NUTS-3 regions in the European Alps (the NUTS-3 region 'CH022'
of Freiburg region in Switzerland reaching -1.92 at 900 m), as well as the coast of Sweden and in the South of the domain.

### 3.2.2 Bias and correlation

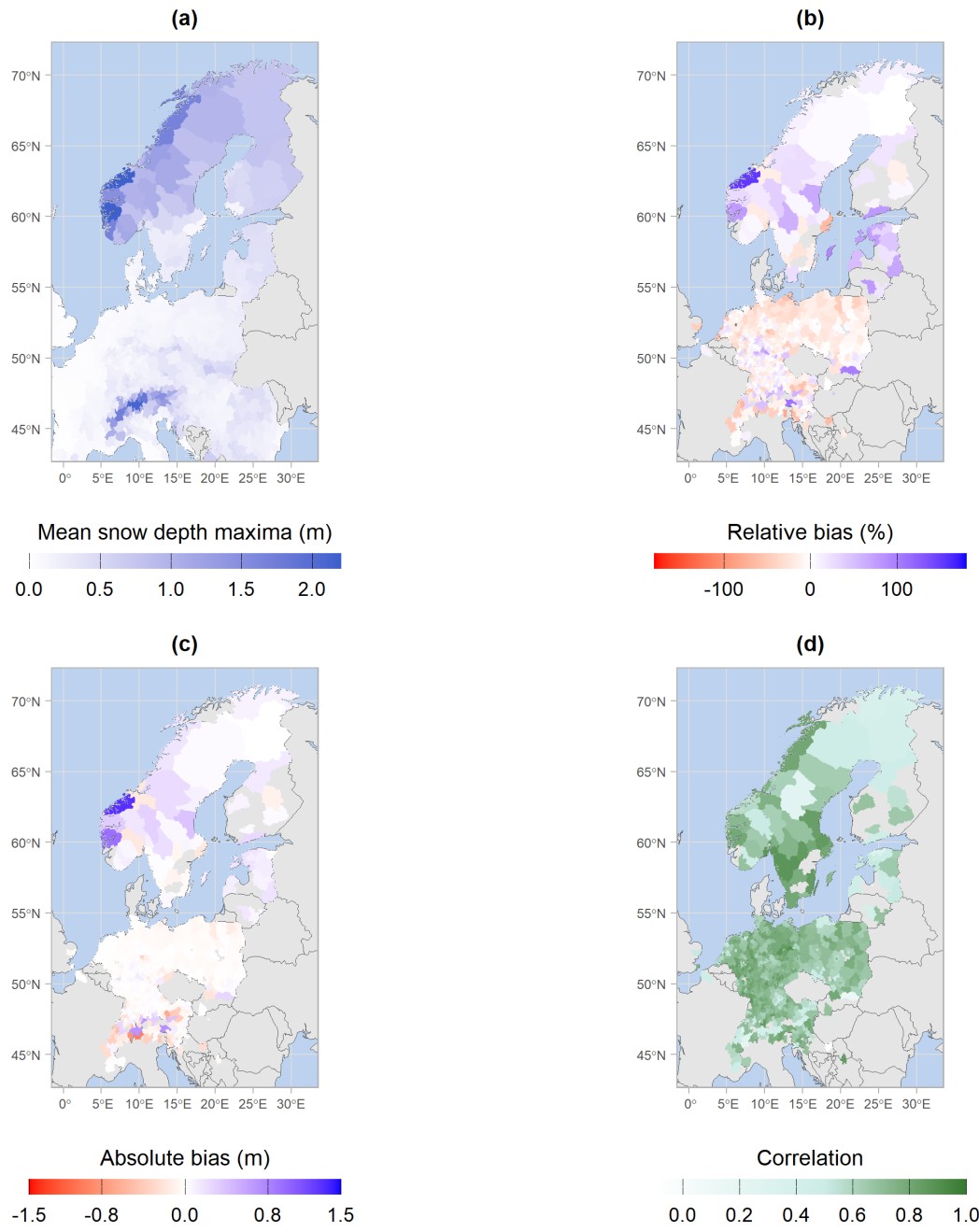

**Figure 5.** MTMSI mean snow depth maxima (a), MTMSI evaluation scores of relative (b) and absolute (c) biases, and correlation (d), on each NUTS-3 region's mean elevation.

Figure 5 shows the mean of MTMSI snow depth maxima at the mean elevation of the NUTS-3 regions, along with the evaluation scores at the mean elevation of the NUTS-3 regions, in a similar way to Figure 4. Figure 5a shows that the highest mean snow depth maxima are obtained for NUTS-3 regions located in the Alps and in the Scandinavian region. The relative bias values lie around 0% (Fig. 5b), and 75% of the NUTS-3 regions have relative biases between -33% and 13%. Positive biases are predominant in plain NUTS-3 regions, and negative biases are major in mountain NUTS-3 regions. However, the Italian NUTS-3 region ITH41 reaches the minimum relative bias of -78% at 600 m, and -72% is reached in the Netherlands at 0 m. Large positive values go up to 93% at 0 m in Sweden, and 81% in the Swiss Alps at 2000 m. The outlying value of 225% is reached in the Netherlands too at 0 m and appears in grey as it is out of the color scale. Figure 5c shows that the majority of NUTS-3 regions have small absolute biases at their mean elevation, 50% of the biases being between -0.03 m and 0.01 m and 75% of the biases being between -0.06 m and 0.04 m. The smallest values are reached in the NUTS-3 region where the mean elevations are below 500 m - 'DE724' NUTS-3 region from Hesse in central Germany reaching down to $1.9 \times 10^{-4}$ m at 300 m elevation. Outlying positive bias values are highlighted in the coastal NUTS-3 regions of Norway, where bias values reach up to 1.32 m at 600 m elevation. Large biases are also obtained for a few NUTS-3 regions in the European Alps, up to 0.72 m at 1500 m elevation, and down to -1 m at 1800 m elevation.

Figure 5d displays correlation scores at the mean elevation of the NUTS-3 regions. It shows that correlation values are above 0.5 for the majority (94%) of the NUTS, and 75% of the NUTS-3 regions have correlation values above 0.65. Weak correlations are obtained in Latvia (0.09 at 0 m), Slovakia (0.11 at 600 m elevation), in the European Alps (0.20 at 2100 m elevation) and in some areas of the Scandinavian countries (0.25 at 500 m elevation in Sweden).

Overall, the spatial patterns of the KGE scores are in strong agreement with the absolute bias and correlation metrics. Indeed, the highest KGE values are reached where both bias values tend to 0 m and correlation values to 1, and vice versa. Most NUTS-3 regions with low KGE values have both low values of correlation and high bias values, which is expected since the KGE score is composed of these two components and the variability ratio. Figure 5 also highlights higher accuracy of MTMSI in Northern Germany and border countries. Yet, the European Alps and coastal areas of the Mediterranean Sea include cases of highly positive and negative bias values as well as low correlation. Scores obtained in the Scandinavian countries are more patchy, and one coastal NUTS-3 region from Norway has a high bias close to 1.5 m at its mean elevation.

## 3.3 MTMSI dataset skills as a function of elevation in NUTS-3 regions

Figure 6 shows the correlation and absolute bias of the MTMSI annual snow depth maxima, per steps of 500 m elevation ranges, and per latitude range. The boxplots represent the dispersion of these two metrics obtained over the different stations and corresponding NUTS-3/elevation pairs, and result respectively from 1295, 1184, and 9871 scores for latitudes above 55°N, between 48°N and 55°N, and below 48°N.

Correlation values are higher and less dispersed below 1500 m elevation, as 75% of the values are between 0.51 and 0.82 (Fig. 6b). This range of elevation interval is where most evaluations are carried out, as they represent 11,616 evaluations out of a total of 12,350 (see Figure 2d). 1,368 correlation scores are below 0.5 below 1500 m, and can even reach negative values (61

cases). Between 1500 m and 3500 m elevation, correlation values are lower in general and more variable, as 75% are between 0.29 and 0.79.

Below 1500 m, biases are concentrated around 0 m, especially at low latitudes. However, these biases can be large for some NUTS-3 areas (between -2.7 m reached in Norway, to 2 m reached in the Italian Alps). For 249 NUTS-3/elevation pairs, bias scores are out of [-0.5 m; +0.5 m] between 0 m and 1500 m. As the elevation rises, bias values are broader, further away from

zero, more negative than positive (median value of the 1500-2000 m is equal to -0.10 m) - indicating an underestimation of snow depth maxima. As the bias is not normalized, it is expected to reach higher values at large elevations and high latitudes, where larger snowfall amounts and extreme snow depth values generally occur. Furthermore, as indicated above, assimilation of rain gauge measurements in UERRA might also lead to underestimations of snow accumulation at high elevations due to wind-induced undercatch effects (Kochendorfer et al., 2020).

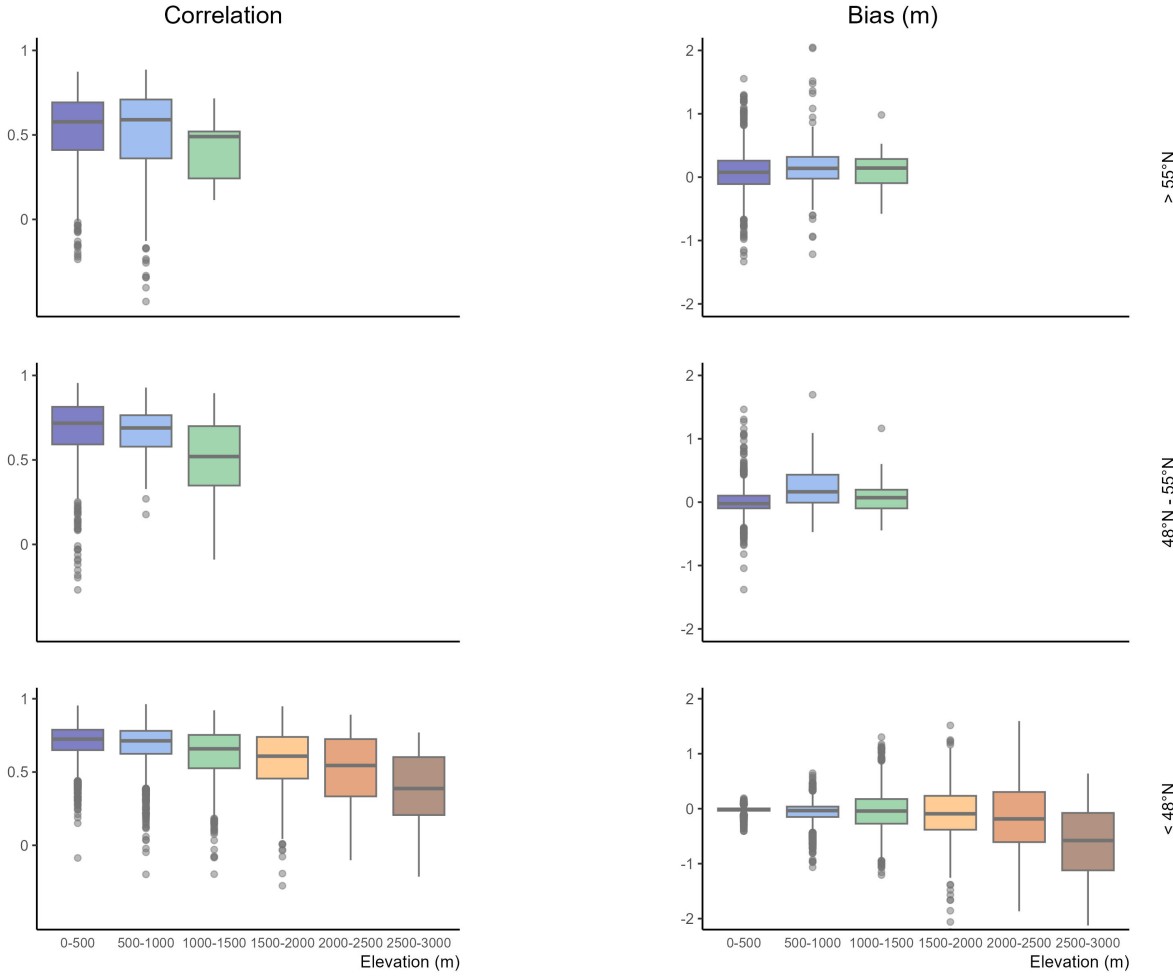

**Figure 6.** Correlation and bias boxplots resulting from MTMSI evaluation by 500 meter elevation bands, the three lines corresponding to different latitude ranges.

**280   3.4   Variability of snow depth maxima observations associated to a given NUTS-3/elevation pair**

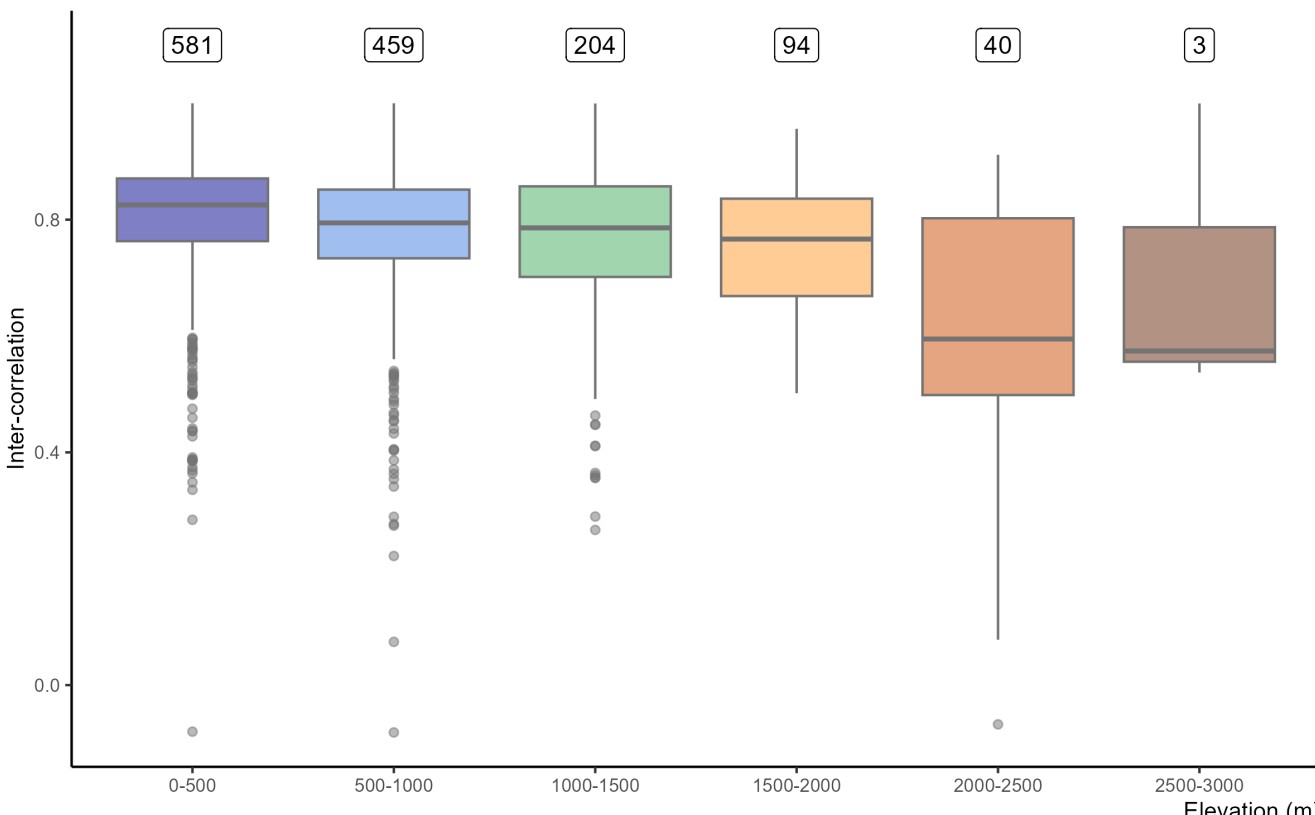

**Figure 7.** Median inter-station correlations for different 500 m elevation bands. The boxplots represent the dispersion of these median inter-station correlations for the different NUTS-3/elevation pairs. The number above each boxplot indicates the number of corresponding NUTS-3/elevation pairs with at least two stations used for the evaluation. At a 2500-3000 m elevation range, the boxplot is obtained using only three median inter-station correlations corresponding to three NUTS-3/elevation pairs and whiskers are extrapolated from these three values.

Figure 7 shows the median correlation of the pool of observations associated with each NUTS-3/elevation pair, when more than one station was used for the evaluation. We extracted the median of the correlation values between all pairs of stations for each NUTS-3/elevation. These median correlations, obtained solely using the observed maxima, intend to describe the variability of observed maxima inside a NUTS-3 region, for a given elevation range, due to local meteorological effects. Median inter-station
correlations are higher and less dispersed below 1500 m elevation, as 75% of the values are contained between 0.66 and 0.89, and 10% are below 0.5 for this elevation range. These scores are more spread out for the 2000-2500 m elevation range, as 50% of the values are contained between 0.51 and 0.80. This suggests lower agreement between stations within this elevation range.

When the elevation spans from 2500 to 3000 m, values are more gathered around 0.75, but it is based on only nine stations providing median inter-station correlations for three NUTS-3/elevation pairs.

## 4 Discussion

### 4.1 Skills of the MTMSI reanalysis dataset for reproducing snow depth extremes on pan-European scale

Our analysis of the MTMSI reanalysis, in terms of snow depth annual maxima, across various climate contexts from the European Alps to Scandinavia, shows that this dataset has an adequate capacity to reproduce snow depth maxima at the scale of NUTS-3 region. Correlations with observed snow depth maxima are above 0.49 for 87.5% of the 12,350 comparisons between observed and simulated snow depth maxima corresponding to the different pairs of NUTS-3 regions and elevations evaluated in this study (Table 2), and greater than 0.59 for 75% of them. Similarly, 75% of the comparisons lead to absolute biases between -0.23 m and 0.15 m. These results can be related to the study of Monteiro and Morin (2023), who found that the monthly snow depth values of MTMSI reanalysis were highly correlated to in-situ observations in the European Alps (around 0.9 at all elevations), and with moderate biases (normalized mean absolute errors smaller than 40% at all elevations). They also highlighted differences smaller than 20% concerning the mean snow season in terms of amplitude and timing of the beginning, peak, and end of the snow season, compared to the in-situ reference.

Since MTMSI does not assimilate snow depth, these results show that the surface analysis through the MESCAN system provided reliable snow cover primary atmospheric drivers (precipitation and temperature), which are then processed by the Crocus model, which includes a fair representation of snow physics (Krinner et al., 2018), able to simulate relevant snow depth and snow water equivalent values for a given meteorological forcing.

We note that while the MTMSI dataset spans a wide range of elevations, the paucity of in-situ observations at higher elevations limits our ability to evaluate the quality of MTMSI at the highest elevation (i.e. above 2500 m). Even if these highest elevation locations are not the most critical for infrastructure design challenges (buildings, roads, railroads, etc.), which are mostly implemented at lower elevations in mountain areas, or in low-lying plain areas, it must be acknowledged that very large snow loads can lead to major hazards at these elevations, and information about extreme snow loads in these areas is critical for the design of defense structures in avalanche-prone areas or mountain huts. We report moderate underestimations at these elevations, potentially due to wind-induced undercatch issues (Kochendorfer et al., 2020). Indeed, UERRA assimilates gauged measurements which probably underestimate solid precipitation at those elevations. In addition, it must be noted that snow depth measurements can also have large errors in complex mountainous terrains (López-Moreno et al., 2013).

In this context, we conclude that the MTMSI dataset provides, in most cases, representative and plausible values of the snow depth time series including annual maximum values, and supports the approach developed by Evin et al. (2025) based on EURO-CORDEX climate projections using the MTMSI reanalysis as a reference for adjusting the climate projections. Evin et al. (2025) provided 50-year return level estimates as a function of global warming levels, based on the historical and future runs of the multi-model ensemble. In addition, the uncertainty of these estimates is assessed using the different runs of the multi-model ensemble and bootstrapping techniques.

## 4.2 Limitations to MTMSI reanalysis data inherited from its specific geometry

The MTMSI reanalysis is designed to represent spatio-temporal variations of meteorological and snow data under past and future climates, based on 0.5% of the UERRA grid points. While this choice has been shown to be efficient overall, some choices may have consequences on the quality and fitness-for-purpose of the dataset depending on the application. The snow depth maxima used in this study are indeed available at the scale of the NUTS-3 regions, which is compatible with other socio-economic indicators. However, this coarse resolution corresponds to purely administrative borders, which implies that they sometimes do not align with the physical geography and local climate zoning (Morin et al., 2021). Still, we found that the stations from similar NUTS-3 regions and elevation are generally highly correlated in terms of their annual snow depth maxima, at least below 1500 m elevation. Elevations above 1500 m relate to mountainous areas where precipitations are more heterogeneous, and for which the NUTS-3 scale could be even less relevant.

The scale is also interfering in the elevation, as at high elevations (e.g. above 1500 m), it has proven sometimes difficult to find NUTS-3 region's nearby pixels with a close elevation from the UERRA reanalysis reference. This can create inconsistencies between neighboring NUTS-3 regions for a given elevation (Evin et al., 2025), as well as inconsistencies in the elevation dependency of the results for a given NUTS-3 region (Morin et al., 2021). This issue is highlighted on the NUTS-3 region 'CH056' in Switzerland (Grisons canton) at the elevation of 2600 m and 2700 m (Figure 8). In 1986, we can see that MTMSI snow depth maxima reaches 3 m at 2600 m, while it reaches 2.37 m at 2700 m. This inconsistency might be explained by the location of the pixels from the UERRA reanalysis reference that were picked to represent this NUTS-3 region. We can note that this inconsistency case is highlighted by the score values, as the simulated snow depth maxima at 2600 m leads to a median correlation of 0.18 and a median bias of -0.63 m, while the simulated snow depth maxima at 2700 m reaches a median correlation of 0.22 and a median bias of -0.85 m.

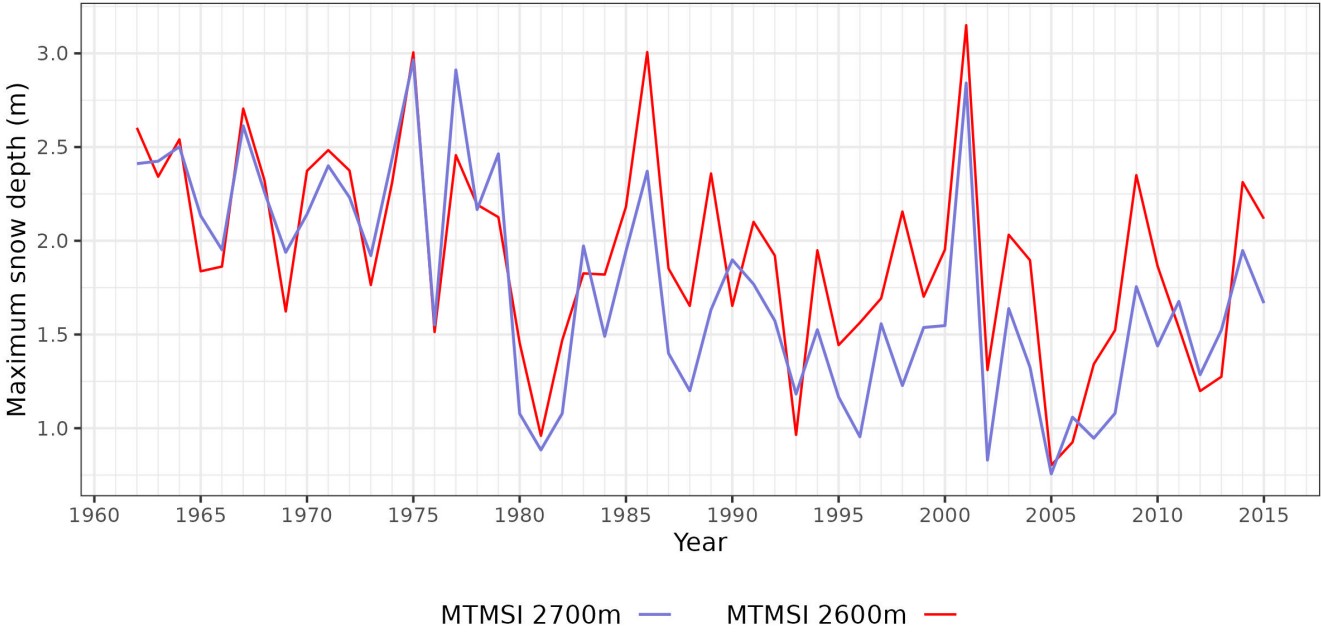

**Figure 8.** Time series of annual snow depth maxima from MTMSI and in-situ stations on NUTS-3 region 'CH056' located in the Swiss Alps (Grisons canton), at 2600 m (red) and 2700 m (blue).

### 4.3 Limitations of the MTMSI reanalysis data inherited from the UERRA MESCAN-SURFEX reanalysis

The MTMSI reanalysis dataset is based on UERRA (UERRA MESCAN-SURFEX), and thus inherits directly from its limitations. Specifically, the quantity and quality of data assimilated into the UERRA vary across the domain. Therefore, heterogeneities in the dataset are likely (https://confluence.ecmwf.int/display/UER/Issues+with+data).

On top of that, Dierickx (2019) pointed out that erroneous observations did enter the UERRA assimilation procedure. As reanalyses are grid based, it is also challenging to correctly reconstruct variables that have large space and time variability, such as precipitation. In particular, low density of precipitation observation networks at high elevation make solid precipitations to be often underestimated. Dierickx (2019) pointed out that in Sweden, UERRA HARMONIE 2m-temperature (which fed UERRA with atmospheric variables) correlations are lowest in the mountain areas (north-west) and along the (east) coast. Figure 9 shows

the time series of annual snow depth maxima from MTMSI and related in-situ observations for the NUTS-3 region 'NO052' (Sogn og Fjordane, Norway) at 0 m and 600 m. At 0 m, MTMSI appears to underestimate snow depth maxima, while at 600 m, it widely overestimates them. This discrepancy can be explained by the heterogeneous orography of this NUTS-3 region, that the reanalysis-based MTMSI is having difficulties to capture. Indeed, the large pool of in-situ observations displayed in Figure 9a shows how important is the variability of the measured values of snow depth throughout this NUTS-3 region.

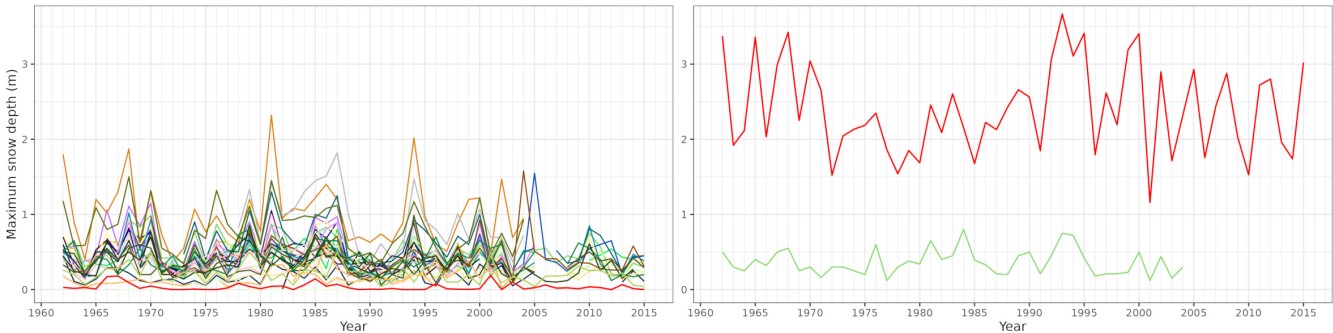

**Figure 9.** Time series of annual snow depth maxima from MTMSI (in red) and in-situ stations (other curves) on NUTS-3 region 'NO052' located on the coast of Norway (Sogn og Fjordane), at 0 m (left) and 600 m (right).

To summarize, results of UERRA in complex terrain, such as mountainous regions or coastal areas, are generally less reliable than results over a more homogeneous terrain. Similarly, Olsson et al. (2023) and van der Schot et al. (2024) highlighted noticeable coastal effects on snowfall on the Finnish and the Greenlandic coasts, respectively.

Nevertheless, compared to larger-scale reanalyses, produced at a horizontal resolution of several tens of km, the UERRA reanalysis at 5.5 km horizontal resolution, and its successor the Copernicus European Reanalysis (CERRA, Ridal et al., 2024) provides more relevant results in complex terrain (Monteiro and Morin, 2023), yet to be improved based on higher-resolution reanalyses in the future, able to better represent meteorological phenomena in mountainous areas, and incorporate more observation data in their data assimilation systems.

## 5 Conclusion

In this study, we have evaluated the MTMSI reanalysis dataset over the time period 1962-2015 regarding the annual snow depth maxima given at NUTS-3 scale. Using multiple in-situ observation datasets, we computed statistical scores to evaluate the MTMSI dataset on 671 NUTS-3 regions, covering a large area stretched from the European Alps to the Scandinavian countries. The evaluation was conducted on a large elevation range as well, from 0 m to 3000 m. In addition to the correlation and the bias, we computed the KGE score, as it enables combining different types of error into a single value. The KGE, which had already been used for assessing reanalysis precipitations, provided useful information in our case. For 75% of the comparisons between observed and simulated snow depth maxima, absolute bias scores between -0.23 m and 0.15 m are reported, as well as correlations and KGE above 0.59 and 0.29, respectively, indicating reasonable skills for most of the NUTS-3/elevation pairs evaluated. We identified some areas where MTMSI did not adequately portray in-situ observations of snow depth maxima, located in the European Alps, and coastal areas of the Netherlands, Norway, Sweden, and Croatia. These poorer results agree with previous studies that evaluated UERRA (from which MTMSI is a subset) and found less correlation in some coastal areas. They can also be explained by the coarse resolution that does not account for coastal effects as well as local climate from mountain areas.

The results of our evaluation of MTMSI, combined with the one conducted by Morin et al. (2021) and Monteiro and Morin (2023), consolidate the estimation of changes in extreme snow load conducted by Evin et al. (2025) on the pan-European territory from MTMSI projections. The estimated changes, harmonized at the European scale, could therefore contribute to updating the Eurocodes standards designed for building safety over the European region. Higher-resolution reanalyses and climate simulations, able to better represent processes conducive to major snowfall events in various environments (coastal, mountainous, etc.), constitute a clear way forward towards improved representation of extremes in snow depth, snow water equivalent, hence snow load extreme values, under past and future climates.

## 6  Code and data availability

All computations were performed with R software version 4.1.2. The codes are available from a repository (GitHub repository: https://github.com/elisakmr/MTMSI-evaluation, last access: 2025-04-11) which includes scripts to perform (1) the extraction of MTMSI and in-situ observation sets; (2) the statistical scores of each NUTS/elevation pair with associated station(s); (3) the figures depicting the data set characteristics and the scores, that are shown in the paper.

The evaluated reanalysis data set is available from a repository (https://doi.org/10.5281/zenodo.15181401) (Kamir et al., 2025). It relates to the dataset underlying the MTMSI data set (Morin et al., 2021), available on the Copernicus Data Store following this url (https://doi.org/10.24381/cds.2fe6a082, last access: 2024-12-02). The dataset collection comprises a netcdf file with the annual snow depth maxima (in meter) at the NUTS-3 scale, a metadata (netcdf) file with the latitude and longitude of the barycenter of the NUTS-3 regions, and the GeoPackage file corresponding to the NUTS-3 regions.

The in-situ observation data set of the European Alps is the one referred to by Matiu et al. (2021), available through https://doi.org/10.5281/zenodo.5109574. The in-situ observation data set of Germany has been provided by the German meteorological service. The in-situ observation data set of Northern and Eastern Europe has been downloaded from https://www.ecad.eu/. The in-situ observation data set of Finland has been provided by the Finnish Meteorological Institute.

*Author contributions.*  The study was designed by EK, SM and GE. EK gathered the data and conducted the analysis with support from SM and GE. PG and BW provided the data from the German meteorological service DWD, and ANA provided the data from the Finnish Meteorological Institute. The draft was written by EK, SM, GE, PG, with contributions from all authors.

*Competing interests.*  The authors declare the following financial interests/personal relationships which may be considered as potential competing interests.

ther geographical representation in this paper. While Copernicus Publications makes every effort to include

appropriate place names, the final responsibility lies with the authors.

*Acknowledgements.* The authors thank Raphaëlle Samacoïts (Météo-France) for computing and providing the dataset of annual maxima of snow depth, and Diego Monteiro (Météo-France - CNRS, CNRM, CEN) for providing the formatted in-situ observation data set from Matiu et al., (2021b).

This work has been performed in the context of the "Framework Partnership Agreement for Copernicus User Uptake" (FPCUP) action

"Estimation of snow load data using Copernicus and in-situ data", which is financed by the European Commission under the FPA no. 275/G/GRO/COPE/17/10042.

We thank Michael Matiu, J. Ignacio López-Moreno and one anonymous reviewer for relevant and useful comments and suggestions on the original manuscript.

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
