# Peer review of "Evaluation of annual maximum snow depth data estimation from the European-wide reanalysis C3S MTMSI (Copernicus Climate Change Service - Mountain Tourism Meteorological and Snow Indicators) against in-situ observations"

_Earth System Science Data, 2025_

## Author Response (AR1)

MORIN Samuel
CNRM, Toulouse, France
samuel.morin@meteo.fr

and EVIN Guillaume
IGE, Grenoble, France
Guillaume.evin@inrae.fr

Toulouse, October 17 2025

Revision of the manuscript Kamir et al. (2025)

Dear Editor,

We are grateful to the reviewers for the feedback provided on the discussion version of our manuscript. We have addressed all comments and are happy to provide, below, a point-by-point reply to the issues and comments raised by the reviewers. We provide a revised manuscript as well as a track-change version of the manuscript.

We hope that this revised version will be deemed sufficient for acceptance and publication. We remain available for further interaction if need be.

Sincerely yours,

Samuel Morin and Guillaume Evin, on behalf of the author team

**Reviewer #1**

RC#1.1. Kamir et al. evaluate maximum snow depth from MTMSI, a reanalysis-based snow indicator dataset over NUTS regions, with respect to in-situ snow depth observations. The paper has high quality graphics and is well readable. Their analysis is able to guide extreme snow loads identification and future assessments, which is of societal relevance. The study has a few strengths. The evaluation covers a large geographic area in Europe, from the Mediterranean to Scandinavia. The study is interesting and worthy of publication because it looks at extreme values, while most previous studies only look at averages. Since the behavior can be different comparing means or extremes, it is highly relevant, also because impacts are much higher for extremes than means.

We thank the reviewer for this positive feedback and for acknowledging the specificity of this study which focuses on snow extremes.

RC#1.2. However, also a few issues as outlined below. Finally, I'm not sure if the study lies within the scope of ESSD, since the authors do not produce/describe a dataset, but instead just extract values from an existing one. But this is for the editor to decide. Also please disregard the formal manuscript rating in the editorial system, since the points there refer to a novel dataset, and thus 2) significance and 4) data quality cannot be meaningfully answered for this study.

It is true that the simulations were produced during a previous study. However, the raw simulations (daily snow depth) were not available and the dataset shared and evaluated in

this study (i.e. annual maxima of daily snow depth and snow water equivalent) has not been published before. As such, we consider that snow maxima provided in this study can be considered as new since it was not accessible before.

RC#1.3. While the evaluation of extremes is relevant, the analysis performed in the study is at times superficial. I acknowledge the complicated structure of the MTMSI dataset, which makes it challenging. The elevational analysis, for example, is challenging to understand in the current form. It is unclear how many stations/regions at which elevations in which locations were used/considered. Maybe it would be useful to distinguish between plain and mountain NUTS. Also the analysis needs to somehow consider the latitudinal gradient.

We agree that the presentation of the results could be improved and clarified concerning the dependence to the elevation. We have modified and complemented some figures:

- We now present three different maps in Figure 2. In addition to the number of stations per NUTS-3 area, we also present the number of elevation bands which have been evaluated in the MTMSI dataset and the average number of stations per elevation band participating in the evaluation.
- Concerning the latitudinal gradient, Figure 6 of the revised manuscript shows the biases and correlations according to the elevation and different altitudinal intervals: less than 48° ; between 48° and 55° ; more than 55°.

RC#1.4. Station data are used multiple times for the different NUTS/elevation groups. While I understand the author's needs to cover as much as possible of the MTMSI dataset, this still feels like inflating the analysis or the number of observation pairs. There is some discussion at the end, but only based on one example. This might deserve some more thinking.

We have performed an additional experiment where only stations within the elevation band are included in the evaluation, which avoids this inflation. This figure is provided in the Supplement (Fig. S1) and replicates Figure 5 of the revised manuscript. The overall patterns are very similar. Some differences can be noticed. A highly positive bias of a Norwegian coastal NUTS-3 region is absent as no station could be selected. However, some of the important absolute biases of Alpine NUTS-3 regions are larger.

[Figure]

(a)

Mean snow depth maxima (m)

0.0    0.5    1.0    1.5    2.0

(b)

Relative bias (%)

-100    0    100

(c)

Absolute bias (m)

-1.5    -0.8    0.0    0.8    1.5

(d)

Correlation

0.0    0.2    0.4    0.6    0.8    1.0

**RC#1.5. Negative bias at high elevations (Fig 6) is not discussed in detail.**

We have added this sentence: "Furthermore, as indicated above, assimilation of rain gauge measurements in UERRA might also lead to underestimations of snow accumulation at high elevations due to wind-induced undercatch effects (Kochendorfer et al., 2020)."

**RC#1.6. Some comparison of extremes versus means would be highly beneficial. Also to put the previous studies in context. While this could be done just discussing the numbers of previous studies with the ones here, alternatively the analysis, or parts thereof, could be repeated for means.**

The analysis for annual snow depth means has already been performed in Monteiro and Morin (2023, The Cryosphere). For the sake of brevity, we do not repeat this analysis in the current study, although we fully agree that a joint analysis of annual mean / maximum snow depth would be interesting per se.

RC#1.7. L11 "satisfactory" means? Some numbers would be helpful.

We agree that "satisfactory" is ambiguous and we have added quantitative results (bias, correlation and KGE scores) in the abstract and in the conclusion.

RC#1.8. L29, this paragraph is more a description of methods, not introduction.

Thanks for this comment. We agree that it was redundant with the description of the MTMSI dataset and we have removed most of this paragraph.

RC#1.9. L53 Monteiro and Morin did not use remotely sensed snow depth, only in-situ snow depth (and remote sensing snow cover fraction)

We agree that this part was misleading. We have replaced "remote sensing snow depth" by "remote sensing snow cover fraction".

RC#1.9. L55 again, "satisfying" is vague. … I see you use this phrase a lot, if possible please add a quantitative number for clearness.

We have added the following sentence: "Monthly snow depth values lead to mean absolute errors generally below 30%, and correlation values close to 0.9 at all elevations."

RC#1.10. L78 based on what criteria were plain and mountain NUTS3 distinguished?

The plain and mountain NUTS3 distinction was based according to their assigned ski tourism character (see Morin et al. 2021, https://doi.org/10.1016/j.cliser.2021.100215), namely "a combination of geographical information system approaches with expert analysis of regions hosting significant ski tourism across Europe".

RC#1.11. Why did you not use the NH-SWE dataset, which is basically ECAD with quality checks. https://essd.copernicus.org/articles/15/2577/2023/

We used a subset of 5877 stations from ECA&D. However, as shown on the map below, the dataset compiled by Fontrodona-Bach et al. provides a rather scarce data coverage for most of Europe. We have thus exploited a denser snow depth observational dataset from the European Alps region, complemented by densest-possible country specific observations from Germany and Finland, thereby providing data over a north-south gradient across most of Europe.

[Figure]

RC#1.12. Related: The study needs some explanation/discussion why there have not been used tools such as the delta_snow model or HS2SWE, which convert daily time series of snow depth to SWE and vice versa without further input. Snow depth is a good proxy for SWE, but still, the sensitivity of results to the chosen approach could use some further analysis and/or discussion.

The choice was to evaluate the direct measurements and not processed measurements. Indeed, delta_snow model or HS2SWE could potentially affect the evaluations if they are not accurate at some locations. While we agree that the relationship between snow depth and SWE is not straightforward, previous evaluations using Crocus show that there are no major differences of performances between snow depth and SWE simulations (see l.48-51 of the revised manuscript).

RC#1.13. L178-181 Unclear, please reformulate or expand.

We agree that this paragraph was unclear. We have reformulated it.

RC#1.14. Why did you not consider a measure for spread, like RMSE or MAE?

This is a good suggestion, and we provide the MAE in the Supplement (Fig. S2).

[Figure]

MAE (m)   MAE > 2.5m

0.0   0.4   0.8   1.2   1.6

RC#1.15. Besides absolute bias, I recommend investigating also the relative bias, which is often a more useful metric for zero-bounded variables.

We thank the reviewer for this suggestion. Figure 5b of the revised manuscript presents the relative biases.

RC#1.16. Since you are looking at extremes, why did you not consider extreme value theory, or metrics based on GEV distributions, return levels, or similar?

Thank you for this suggestion. In this study, we are interested in the accurate reproduction of the maxima, and their chronology, i.e. we wanted to check if the largest maxima occur at the same year and with the same magnitude in observed and simulated time series. Extreme value theory, on the other side, only provides an evaluation in terms of distribution, and can lead to uncertain results in extrapolation (see Klemes, 2000a,b). An additional difficulty is the fitting of the GEV distribution which can be challenging with small samples (i.e. 20-30 maxima).

RC#1.17. Fig 5a, better if you switch red and blue colours, for easier visual perception (red drier, blue wetter)

We agree and we have modified the color scale in the revised manuscript.

RC#1.18. L227 you mean lower instead of larger?

We mean larger than most of the biases, but we agree that it was confusing considering the previous sentence. We have replaced "larger" by "large".

RC#1.19. For a better understanding from your readership, I suggest adding a map of the mean maxima over the NUTS regions, so the readers can also put the bias values in perspective. In addition to also showing the relative bias (see comment above).

Thank you for this suggestion. Figure 5a of the revised manuscript provides a map showing the mean snow depth maxima.

RC#1.20. Fig 6: are there spatial pattern to this? I would recommend to split at least by latitude bands, since 1000m in the Alps is very different to 1000m in north Scandinavia.

See our response to comment: RC#1.3.

RC#1.21. Figure 7 would be better suited further up, even in methods or beginning of results, to explain the approach.

As suggested, we have moved this figure and its interpretation at the beginning of the section "Results".

RC#1.22. Sec 3.3 unclear; also why only correlation is shown and not bias. Is this related to regions > 2 stations?

Thank you for this comment. We have modified this paragraph and added the number of station pairs in the figure. Please note that this figure does not intend to evaluate the accuracy of MTMSI, but only the dispersion of observed maxima among the different stations, for a given NUTS-3 region and a given elevation band. Therefore, bias values cannot be shown on this figure.

RC#1.23. Sec 4.1: Vague, since MTMSI has already been evaluated on similar but slightly different variables, please put your results in more quantitative comparison.

We thank the reviewer for this comment. We agree that adding quantitative results strengthens the conclusions and we have added different quantitative results in this subsection.

References

Klemeš, V. « Tall Tales about Tails of Hydrological Distributions. I. » *Journal of Hydrologic Engineering* 5, nº 3 (2000a): 227-31. https://doi.org/10.1061/(ASCE)1084-0699(2000)5:3(227).

Klemeš, V. « Tall Tales about Tails of Hydrological Distributions. II. » *Journal of Hydrologic Engineering* 5, nº 3 (2000b): 232-39. https://doi.org/10.1061/(ASCE)1084-0699(2000)5:3(232).

**Reviewer #2**

RC#2.1. The manuscript presents the validation of annual maxima snow depth from a snow product covering large parts of Europe. The manuscript is well-written, the methodology is clear and well-suited to the purpose of the study, and the limitations of the dataset are honestly discussed. Even if the data set shows evident problems at high elevation, and only annual maxima is a snow parameter a bit limited for many research and applied uses, all data about snow is welcome for the community, especially when covering a very large area of Europe. Thus, I recommend the publication of the work in ESSD after considering a few changes to be considered for a revised version.

We thank the reviewer for this positive opinion about this study, and for these constructive comments.

RC#2.2. If not the title, the abstract should show the period covered by the dataset.

Thanks for this suggestion. We have added the information about the period covered by the evaluation (1962-2015) in the abstract.

RC#2.3. In methods, I wonder if it would be more interesting to use a relative bias than the bias itself (or present both). In part, the relative low bias at low elevation can be explained by low snow depth values. If bias is divided by the average value, and expressed in % or 0-1 units it could provide a better representation of the validity of the dataset.

This suggestion was also made by reviewer #1 and the revised Figure 5b shows relative biases.

RC#2.4. Along the results section there are many qualitative references about the accuracy estimators instead of giving the values or ranges of values (to say that a r value of 0.6 is "satisfactory" is very relative..).

We thank the reviewer for this comment that was also raised by the other reviewer. We have added many quantitative results in the abstract, in the results section and in the discussion. We have also added Table 2 in section 3.2 which provides a quantitative summary of the evaluation scores.

RC#2.5. Figure 2 shows stations in Catalonia, but is not mentioned anymore. At some point is mentioned Andorra (small but relevant in terms of snow and snow loads), but it is not shown in the figures (i.e. could be zoomed in figures 4 and 5).

Figure 2 shows all the stations providing snow depth maxima but some of them were discarded in the evaluation because they provide less than 20 snow depth annual maxima or because they do not match the elevation band which has been simulated in the MTMSI dataset.

RC#2.6. In discussion, I have the feeling that large errors at high elevation are softened arguing that most of the constructions are at low elevation. But many critical infrastructures are also at high elevations and are where snow loads often represent a big problem. Just to mild the assessment.

Thank you for this comment. We have adjusted the text to acknowledge that large snow loads can occur at very high elevations and can lead to major hazards.

**Reviewer #3**

**RC3.1. The paper evaluates an existing annual maxima snow depth product, from the C3S MTMSI, against in situ snow depth observations to ensure its validity, going a step further than in previous assessments. The original dataset appears to be used for various purposes, including infrastructure design, and therefore, I find the comparison carried out extremely useful to ensure that the extreme snow values used are trustworthy for this purpose. In addition, the study covers a large range of latitudes within Europe, providing the evaluation with an additional strength.**

We thank the reviewer for the positive feedback.

**RC3.2. I am not sure that the paper fits in the scope of the journal since it does not present a new dataset but rather carries out an evaluation over an existing one.**

Thank you for this comment. This is an important point that must be clarified. Indeed, the MTMSI dataset provides 39 indicators characterizing atmospheric and snow conditions and is primarily dedicated to mountain tourism stakeholders. The dataset shared and evaluated in this study provides annual maxima of daily snow depth and snow water equivalent and was not available before. As such, we consider that snow maxima provided in this study can be considered as new since it was not accessible before.

**RC3.3. I miss in the abstract a reference to the length of the dataset.**

We have added this information in the abstract.

**RC3.4. Additionally, the reader would appreciate a brief description of the NUTS-3 spatial resolution. Personally, after reading the manuscript, I realized that it is an administrative regionalization, but it was unclear to me before seeing Figure 1.**

We are not sure to understand what is missing here. In the introduction, it was indicated in the second paragraph that "NUTS-3 regions [...] correspond to administratively relevant regions defined for each country across Europe" and it was reminded in Section 2.2.1 that "The primary geographical unit components of the MTMSI dataset are NUTS-3 regions, corresponding to administratively relevant regions defined for each country across Europe". Following the reviewer's comment, we have added this information when introducing Figure 1 too.

**RC3.5. The methodology used by Morin et al. (2021) to derive C3S MTMSI is explained twice, both in the introduction and the method section. I think this is repetitive.**

We agree and this point was also raised by reviewer #1 (see comment RC#1.8.). We have removed this information from the introduction.

**RC3.6. Could you elaborate a bit more about the error/uncertainty you are assuming when choosing just a single annual value from each in situ station?**

We are not sure if we understand this comment. Annual maxima are a standard indicator of extreme values in geosciences, as it extracts the largest values of a dataset and is related to interesting statistical properties (extreme value theory).

**RC3.7. The way in which the annual maxima snow depth pairs are chosen for representing the elevation bands in the NUTS-3 areas is not fully clear. What happens if there is more than one situ observation per elevation band? And the opposite, only one observation per NUTS-3 region in a mountainous site?**

We have reworked Section 2.3.2., with the intent of making it clearer. When there are more than one station for an elevation band, we compute median scores among all stations (median bias, median correlation, median KGE). This is now illustrated at the beginning of the section Results in Figure 3 (which was previously at the end of the section). If there is only one station in a mountainous site, only elevation bands within +/- 150 m of the elevation of the station are evaluated.

**RC3.8. The representation of the results of the assessment of annual maxima snow depth by elevation band are not sufficiently clear. I understand that it is difficult to spatially show these results, which are a combination of both NUTS-3 and elevation. I suggest to show the effect of the elevation band on the annual maxima snow depth performance by mountain range/area. I understand high elevations are mainly located in the Alps, but do they have the same performance in low elevation from Germany, the Netherlands, or Scandinavia?**

Thank you for this comment. As suggested by reviewer #1 (see our response to comment RC#1.3.), we have adapted Figure 6 according to the latitudes, which roughly cover three large regions:

- >55°N: Scandinavia,

- [48°N, 55°N]: Germany, Netherlands,

- <48°N: Alps.

**RC3.9. KGE is a nice metric to evaluate high values in time series. In general, -0.41 can be considered the threshold of acceptable performance, as you have highlighted in Figure 4. However, since the higher the KGE, the better the performance, I would recommend using a more contrasted color ramp palette to understand exactly which KGE values correspond to each of the blues. In addition, since KGE is a metric composed of three other metrics, I would explore the performance using these three other metrics. You are already using the correlation coefficient, but not the relative errors (beta in the manuscript KGE equation) and the relative error on the deviation (gamma, in the manuscript KGE equation). I would recommend using them rather than the absolute error.**

Thank you for the suggestion about the color scale in Figure 4. We agree that it was difficult to distinguish different KGE performances and we have modified the colors to highlight differences, using the following intervals for KGE values: [-3,-2,-1,-0.41,0,0.5,1]. Concerning the comment about the other metrics, and as suggested by the other reviewers, we propose to add the relative bias as another criteria. The relative error on the deviation was showing very similar patterns than the KGE and was deemed redundant.

[Figure]

#RC3.10. In the discussion section, I would comment further the bias found at the higher elevation and also include a sentence referring to the error made by the in situ snow depth measurements.

We have revised this part of the discussion according to these comments (end of section 4.1). In particular, we have added the following sentence: "In addition, it must be noted that snow depth measurements can also have large errors in complex mountainous terrains (López-Moreno et al., 2013)."